# *Zbtb40* Deficiency Leads to Morphological and Phenotypic Abnormalities of Spermatocytes and Spermatozoa and Causes Male Infertility

**DOI:** 10.3390/cells12091264

**Published:** 2023-04-26

**Authors:** Yinghong Cui, Mingqing Zhou, Quanyuan He, Zuping He

**Affiliations:** 1The Key Laboratory of Model Animals and Stem Cell Biology in Hunan Province, Hunan Normal University School of Medicine, Changsha 410013, China; 2The Research Center of Reproduction and Translational Medicine of Hunan Province, Changsha 410013, China; 3The Manufacture-Based Learning & Research Demonstration Center for Human Reproductive Health New Technology of Hunan Normal University, Changsha 410013, China

**Keywords:** *Zbtb40* knockout, ZBTB40 mutation, spermatocytes, spermatids, telomere length, male infertility

## Abstract

Studies on the gene regulation of spermatogenesis are of unusual significance for maintaining male reproduction and treating male infertility. Here, we have demonstrated, for the first time, that a loss of ZBTB40 function leads to abnormalities in the morphological and phenotypic characteristics of mouse spermatocytes and spermatids as well as male infertility. We revealed that Zbtb40 was expressed in spermatocytes of mouse testes, and it was co-localized with γH2AX in mouse secondary spermatocytes. Interestingly, spermatocytes of *Zbtb40* knockout mice had longer telomeres, compromised double-strand break (DSB) repair in the sex chromosome, and a higher apoptosis ratio compared to wild-type (WT) mice. The testis weight, testicular volume, and cauda epididymis body weight of the *Zbtb40*^+/−^ male mice were significantly lower than in WT mice. Mating tests indicated that *Zbtb40*^+/−^ male mice were able to mate normally, but they failed to produce any pups. Notably, sperm of *Zbtb40*^+/−^ mice showed flagellum deformities and abnormal acrosome biogenesis. Furthermore, a ZBTB40 mutation was associated with non-obstructive azoospermia. Our results implicate that ZBTB40 deficiency leads to morphological and phenotypic abnormalities of spermatocytes and spermatids and causes male infertility. This study thus offers a new genetic mechanism regulating mammalian spermatogenesis and provides a novel target for gene therapy in male infertility.

## 1. Introduction

Spermatogenesis is strictly organized and precisely regulated by genetic and epigenetic factors in mammals. It is composed of three main processes, including mitosis of spermatogonial stem cells (SSCs) [1], meiosis of spermatocytes to generate round spermatids [2], and morphological and biochemical changes of round spermatids to become mature spermatozoa [2]. Abnormal spermatogenesis can result in male infertility that affects about 15% of couples worldwide [3]. Non-obstructive azoospermia (NOA) is one of the most severe types of male infertility, which has been defined as the absence of spermatids in the ejaculate without obstructive factors, and it includes azoospermia, asthenozoospermia, teratozoospermia, and oligozoospermia. NOA can be caused by congenital causes, such as Kallmann syndrome and Y chromosome microdeletion. There are also some idiopathic NOA patients, accounting for about 30–50% of male infertility cases. To elucidate the pathogenesis and seek potential therapeutic targets in NOA are of great significance for treating male infertility patients. Over the past decades, a number of genes have been identified to be associated with NOA. Briefly, mutations in *DNAH10* [4], DNAH17 [5], *BRWD1* [6], CFAP47 [7], and *CFAP65* [8] have been demonstrated to cause asthenoteratospermia. A RAD51AP2 mutation has been found in patients with unaligned chromosomes in spermatocytes, and its mutation in mice results in the decreased crossover formation between sex chromosomes [9]. Nevertheless, the associations between gene knockouts or mutations and NOA remain largely unknown.

Zinc finger and BTB domain 40 (ZBTB40) is a member of the ZBTB family, a group of transcriptional regulators that are evolutionarily conserved [10]. These proteins feature several C-terminal C2H2/Krüppel-type zinc finger (ZF) domains and an N-terminal BTB (broad-complex, tram-track, and bric-a-brac) domain [11]. The main function of the ZF domain has been reported to mediate DNA binding, while its BTB domain promotes protein–protein interaction [12]. As such, the domain of ZF mediates chromatin remodeling, gene expression, and protein stability. It has been shown that ZBTB proteins are involved in DNA damage responses and cell-cycle progressions [11], and ZBTB16, also known as PLZF, has been identified as a hallmark for the self-renewal of mouse SSCs [13]. Recent studies have reported that *Zbtb40* is likely a candidate gene for mineralized nodule formation, suggesting that ZBTB40 may be a new target for future treatment of osteoporosis [14]. Interestingly, the expression levels of ZBTB family genes were significantly different in male germ cells of *Epinephelus coioides* [14]. Further analysis indicated that *Zbtb40* is only expressed in male germ cells, reflecting that *Zbtb40* may participate in regulating spermatogenesis [14]. However, the functions of ZBTB40 in mediating mammalian spermatogenesis, including mitosis, meiosis, and spermiogenesis, are unclear. The aim of this study was to explore the expression and role of ZBTB40 in mediating male reproduction and male fertility. To this end, we hypothesized that the loss of ZBTB40 function may result in abnormal spermatogenesis and male infertility.

## 2. Results

### 2.1. ZBTB40 Is Expressed in Mouse Spermatocytes

It has been reported that ZBTB40 is essential for spermatogenesis in *Epinephelus coioides* [15]. To explore the function of ZBTB40 in regulating mammalian spermatogenesis, we detected the expression of ZBTB40 in mouse testes using immunohistochemistry. We found that ZBTB40 was specifically present in mouse spermatocytes (Figure 1A). Interestingly, most of the ZBTB40 foci were co-localized with γH2AX (Figure 1B), a hallmark of chromatin remodeling and inactivation of sex chromosome events during spermatogenesis. These results raised the possibility that ZBTB40 may be involved in mediating mammalian spermatogenesis.

### 2.2. Zbtb40^+/−^ Male Mice Exhibit Infertility

To investigate the physiological function of Zbtb40, we generated the *Zbtb40* KO mice (−400bp or −391bp, Appendix A) using CRISPR/Cas9 gene-editing technology. The KO efficiency of Zbtb40 at the mRNA and protein level in mouse testes was evaluated and verified via PCR (Appendix A), Western blots (Appendix A), and immunohistochemical staining (Appendix A).

These two types of *Zbtb40* KO mice had similar body sizes and coat colors as WT mice (Appendix A). *Zbtb40*^−/−^(−391bp) mice could produce offspring normally, and there was no significant difference between their number of offspring and that of WT mice (Appendix A). Significantly, *Zbtb40* KO (−400bp) male mice were infertile, and female mice (−400bp) showed decreased fertility. When *Zbtb40*^+/−^(−400bp) male mice were mated with *Zbtb40*^+/−^(−400bp) or wild-type female mice, they were unable to produce any pups.

We revealed that testis size (Figure 2A, left panel), cauda epididymis (Figure 2A, right panel), testicular volume (117.7 ± 4.055 vs. 86.03 ± 7.151, *p* = 0.0183, Figure 2B), the cauda epididymis and body weight ratio (2.720 ± 0.1825 vs. 1.732 ± 0.1506, *p* = 0.0140, Figure 2C), and the testis/body weight ratio (6.247 ± 0.4114 vs. 3.477 ± 0.4742, *p* = 0.0116, Figure 2D) of the 5-month-old *Zbtb40*^+/−^ male mice were significantly lower than the WT mice. The number of spermatozoa in the cauda epididymis from the *Zbtb40*^+/−^ mice was remarkably reduced when compared to the WT mice (28.33 ± 2.906 vs. 14.30 ± 1.531, *p* = 0.0129, Figure 2E), and more than 95% of the spermatozoa of the *Zbtb40*^+/−^ mice were immotile (80.00 ± 2.581 vs. 6.967 ± 2.236, *p* < 0.0001, Figure 2F). Histological analysis showed that there was a substantial decline of male germ cells in the testes of *Zbtb40*^+/−^ mice compared with the WT mice (Figure 2G, upper panel), while no spermatozoa were observed in the cauda epididymis of the *Zbtb40*^+/−^ mice (Figure 2G, lower panel). Importantly, we found pyknotic spermatocytes with characteristics of apoptosis and mitotic spermatogonia (Figure 2H, red arrows).

### 2.3. Zbtb40 KO Causes Apoptosis and Inhibits the Proliferation of Male Germ Cells

We next probed the function of Zbtb40 in controlling apoptosis and the proliferation of male germ cells. The TUNEL assay showed an increase in apoptotic male germ cells in *Zbtb40*^+/−^ testes compared with those of WT mice (2.600 ± 0.9747 vs. 39.93 ± 4.299, *p* = 0.0001, Figure 3A,B). Furthermore, the percentage of Ki67-positive germ cells was significantly reduced in the *Zbtb40*^+/−^ testes (74.03 ± 4.240 vs. 26.37 ± 4.707, *p* = 0.0017, Figure 3C,D). These results indicate that Zbtb40 deficiency leads to the enhancement of apoptosis and the inhibition of the proliferation of male germ cells.

### 2.4. Zbtb40^+/−^ Affects Spermatocyte Recombination and Pairing of the Sex Chromosomes

We detected that meiosis of the spreads of spermatocytes via immunofluorescence staining for synaptonemal complex protein 3 (SYCP3), the lateral element component, and γH2AX, a marker of double-strand breaks (DSBs). Significantly, the percentages of pachytene spermatocytes (10.50 ± 0.8145, *n* = 869 vs. 21.20 ± 2.821, *n* = 730, *p* = 0.0219) and diplotene spermatocytes (15.10 ± 1.550, *n* = 869 vs. 41.50 ± 5.838, *n* = 730, *p* = 0.0120) with univalent XY in the *Zbtb40*^+/−^ mice were enhanced when compared to the WT mice (Figure 4A,B). Unlike autosomes, there is a very short but highly homologous DNA sequence, namely the pseudoautosomal region (PAR), at the ends of the long arms and the distal ends of the short arms of the human X and Y chromosomes, where meiotic pairing and chromosome interchange occur [16]. This result suggests that the *Zbtb40* knockout renders XY chromosomes more susceptible to precocious separation and affects the recombination and pairing of the sex chromosomes in meiotic prophase. No significant difference was seen in the percentages of leptotene, zygotene, pachytene, or diplotene spermatocytes between *Zbtb40*^+/−^ mice and the WT mice (Appendix A).

### 2.5. The Spermatozoa of Zbtb40^+/−^ Male Mice Showed Flagellum Deformities and Abnormal Acrosome Biogenesis

Since the number of motive sperm in the cauda epididymis was significantly reduced, we then observed the changes in sperm morphology and structure. In comparison with WT mice, the number of *Zbtb40^+/^*^−^ mouse sperm was significantly reduced with severe morphological defects (Figure 5A–C), including acrosome deletion and multiple flagellum deformities (bent, curved, or absent sperm flagella) (Figure 5A−C). We used a transmission electric microscope (TEM) to further analyze the flagellum ultrastructure in the *Zbtb40*^+/−^ mouse spermatozoa. The cross-sections of sperm flagella of *Zbtb40*^+/−^ mice showed the disorganized axoneme and peri-axoneme structures, e.g., the absence or disorganization of doublet microtubules (DMT) and the central pair complex (CPC) (Figure 5D). An abnormal mitochondrial sheath (MS) in the midpiece was observed in the *Zbtb40*^+/−^ mice (Figure 5D). Additionally, impaired flagella formation in the *Zbtb40*^+/−^ mice was also visible by immunofluorescence analyses using an α-tubulin antibody. As shown in Figure 5E, the ratio of absent and malformed α-tubulin in the principal piece of sperm tail in the *Zbtb40*^+/−^ mice was significantly higher than that in the WT mice (66.33 ± 10.84 vs. 18.0 ± 2.309, *p* = 0.0120, Figure 5E,F).

Using TEM, we also observed abnormal acrosome biogenesis in the *Zbtb40*^+/−^ testes (Figure 6A). To further clarify the integrity of *Zbtb40*^+/−^ mouse sperm, the acrosomes were labeled with fluorescein isothiocyanate-Pisum sativum agglutinin (FITC-PSA), and notably, the percentage of acrosome-reacted spermatozoa was higher in the *Zbtb40^+/−−^* mice than in the WT mice (81.10 ± 44.050 vs. 25.77 ± 11.99, *p* = 0.0120, Figure 6B). Collectively, these results indicate that *Zbtb40*^+/−^ leads to various defects of spermatozoa in vivo, including abnormal ultrastructure and acrosome biogenesis.

### 2.6. Zbtb40^+/−^ Mice Have Longer Telomeres in Spermatocytes

We have previously downloaded 377 ChIP-Seq data of 127 nuclear proteins and 8 histone markers in the K562 cells and calculated the ratios of repetitive reads to the total aligned reads in the ChIP data. We found that ZBTB40 was the top one candidate of telomere-associated proteins (TAPs), which strongly indicates that ZBTB40 is a potential TAP (data not shown). We then asked if Zbtb40 affected mouse spermatogenesis via regulating telomere length. We separated the spermatocytes from mouse testes by STA-PUT (Figure 7A). Interestingly, *Zbtb40*^+/−^ mice had longer telomeres than did the WT mice (5113 ± 380.2, *n* = 232 vs. 7233 ± 291.3, *n* = 480, *p* < 0.0001, Figure 7B,C), indicating that abnormal spermatogenesis in *Zbtb40*^+/−^ testes may result from the regulation of Zbtb40 on telomeres.

### 2.7. ZBTB 40 Mutation Is Associated with Non-Obstructive Azoospermia (NOA)

To determine the clinical relevance of ZBTB40, we analyzed the whole exon sequencing (WES) data from 856 NOA patients. Notably, we identified six men from unrelated Chinese families with *ZBTB40* variants (0.70%, 6/856). One patient had the *ZBTB40* homozygous splicing variant c.1025-8T>-, while two other patients carried the same heterozygous splicing variants c.2833+4T>C. Individuals Y8376, 18D3329867, and 18D3331756 harbored heterozygous missense variants c.25C>G(p.Q9E), c.C3257T(p.T1086M), and c.G3350A(p.G1117E). We further analyzed the pathogenicity of these *ZBTB40* missense variants via SIFT, MutationTaster, PolyPhen-2, and CADD software. All three missense variants were predicted to be damaging and disease-causing, which suggests the important significance of ZBTB40 in NOA.

## 3. Discussion

The prevalence of genetic disorders in male infertility is 15–30%, and they result in the majority of NOA cases [17]. Y chromosome microdeletions, especially the genes located at the AZF (azoospermia factor) region, account for 10–15% of NOA cases and 5–10% of males with severe oligozoospermia [18]. Autosomal genes, e.g., the *CFTR* gene (located on chromosome 7) [19] and the FSH receptor (*FSHR*) gene (located on chromosome 2) [20], have been shown to cause male infertility. In this study, we have revealed that ZBTB40, an autosomal gene located in 1p36.12 in humans, is expressed specifically in mouse spermatocytes, and significantly, loss of *Zbtb40* led to severe azoospermia and multiple morphological abnormalities of sperm. We have demonstrated that the *Zbtb40* knockout mice had longer telomeres and a higher apoptosis ratio. Furthermore, we found several variations of *ZBTB40* in NOA patients. All the *ZBTB40* variants identified in our study were predicted to be damaging and disease-causing. It is of great importance to probe the association between the mutations of the *ZBTB40* gene and male infertility in humans, which could provide novel biomarkers and targets for male infertility diagnosis and gene therapy.

The ZBTB transcription factor family plays an essential role in cell growth, differentiation, and oncogenesis. Several ZBTB proteins, e.g., ZBTB1, ZBTB17, ZBTB7B, and ZBTB27, have been proven to regulate the development and differentiation of conventional T cells [21], while BCL6/ZBTB27 can function as vital proto-oncogenes as a transcriptional repressor [22]. However, the functions of the ZBTB family proteins in controlling spermatogenesis need to be clarified. PLZF, known as ZBTB16, has been shown to be expressed in undifferentiated spermatogonia, which is the first *ZBTB* gene essential for the maintenance of SSCs [23]. Mice lacking *Plzf* undergo a progressive loss of spermatogonia, which indicates that Plzf is important for the self-renewal of SSCs [23]. Mice with the disrupted *Zbtb32* are fertile but at a lower frequency [24], and androgen receptor (AR) signaling is induced in Sertoli cells lacking *Zbtb32* [24]. Zbtb20 is localized specifically in Sertoli cells, and mice with conditional knockout (cKO) *Zbtb20* in Sertoli cells exhibit fertility and have no detectable abnormalities in spermatogenesis [25]. In this study, we found Zbtb40 is specifically located in mouse spermatocytes. We constructed *Zbtb40* knockout mice, and notably, we have demonstrated, for the first time, that Zbtb40 deficiency results in abnormal spermatogenesis and male infertility.

ZBTB40 belongs to the ZBTB protein family, while its function is almost unknown in mediating tissue and organ development. Recently, it has been reported that ZBTB40 has a role in osteogenesis and osteoclastogenesis by regulating bone-metabolism-related gene transcription [26]. Further, a loss of function of Zbtb40 in MC-3T3 cells leads to the reduction of early osteoblast factors that can induce osteoblast lineage and drive osteogenic differentiation [14], suggesting that Zbtb40 may control cell differentiation. During meiosis, cells initiate programmed DNA double-strand breaks (DSB) and repair them by homologous recombination pathways to facilitate the exchange and segregation of homologous chromosomes [27]. As a marker of DSB repair, γH2AX is required for meiosis [28]. We have revealed that Zbtb40 and γH2AX are co-localized in mouse spermatocytes, indicating that Zbtb40 participates in DSB repair of mouse spermatocytes. In *Epinephelus coioides,* Zbtb40 has been found to be expressed in male germ cells, and it is co-localized with Cyp17a1 in spermatogonia and spermatocytes. However, the roles of Zbtb40 in regulating spermatogenesis in *Epinephelus coioides* and in mammals remain unknown [15]. In the present study, we found that a loss of function of ZBTB40 resulted in male infertility. Specifically, *Zbtb40* knockout mice had fewer germ cells and more apoptotic cells in the testes. *Zbtb40* mutation affects the recombination and pairing of the sex chromosomes in meiotic prophase and renders XY chromosomes more susceptible to precocious separation. Moreover, the rates of dysmorphic sperm were increased in the caudal epididymis. Notably, we identified several *ZBTB40* variations in NOA patients, which implies that ZBTN40 is closely related to human spermatogenesis.

Telomeres are special structures at the ends of linear chromosomes. They are important for the stabilization of chromosome end structures and the prevention of inter-chromosome end joining. A recent study has indicated that shortened telomeres induce male germ cell apoptosis [29]. A whole-exome sequencing (WES) for a family in which multiple sons displayed the NOA phenotype identified compound heterozygous frameshift variants of Telomere Repeat-Binding Bouquet Formation Protein 2 (TERB2) [30]. A Terb2 mutation in mice shows defects in the meiotic process and ultimately male infertility [31]. We have found that Zbtb40 is involved in telomere length regulation in alternative lengthening of telomeres (ALT) cells, including the U2OS cells (data not shown). Here we have demonstrated that the spermatocytes of the Zbtb40 mutant mice had longer telomeres than did the WT mice, which suggests that the telomere-related function of Zbtb40 may be essential for mammalian meiosis and spermiogenesis.

In summary, we have found that Zbtb40 is expressed specifically in mouse spermatocytes and that it is co-localized with γH2AX in mouse secondary spermatocytes. Significantly, our loss-of-function study suggests that the *Zbtb40* knockout mice show male infertility, and they assume morphologically defective sperm heads, flagella, and acrosome biogenesis. We have demonstrated, for the first time, that a *Zbtb40* knockout leads to abnormal morphological and phenotypic characteristics of mouse spermatocytes and spermatids and eventual male infertility. Notably, a *Zbtb40* knockout results in longer telomeres of the spermatocytes, compromised DSB repair in sex chromosomes, and a higher apoptosis ratio of male germ cells. We have also identified ZBTB 40 mutations in patients with non-obstructive azoospermia. Collectively, our study indicates that *Zbtb40* deficiency leads to morphological and phenotypic abnormalities of spermatocytes and spermatids, as well as telomere dysfunction, which causes male infertility. This study thus provides a novel insight into genetic regulations of mammalian spermatogenesis, and it lays the basis for gene correction in male infertility therapy.

## 4. Material and Methods

### 4.1. Zbtb40 Knockout Mice

The specific fragment of mouse *Zbtb40* (NM_198248.1) was knocked out using CRISPR/Cas9 gene-editing technology. DNA from the *Zbtb40* knockout mouse tails was extracted and used for PCR with phenotype analysis, while primer sequences of *Zbtb40* genes are shown in Appendix A. Sequencing of *Zbtb40* genes was performed on the selected samples with successful verification. After reproductive cloning, the F1 generation of *Zbtb40* knockout mice was obtained. The testes of adult mice at 8 weeks old were collected, and immunohistochemical staining was performed to compare the expression levels of Zbtb40 in wild-type and *Zbtb40* knockout mice.

### 4.2. Western Blots

Cells were lysed with ice cold NETN (100 mM NaCl, 20 mM Tris-HCl pH8.0, 5 mM EDTA, 0.5% NP-40 with protease inhibitors and DTT). The cell lysates were denatured for 10 min at 100 °C and resolved by 8–16% SDS-PAGE gels and transferred to PVDF membranes (RPN203B, GE life) at 200 mA for 2 h. The membranes were blocked by 5% (*w*/*v*) BSA for 1 h at room temperature (RT), and they were incubated with primary antibodies diluted in QuickBlockTM Western antibody dilution solution (P0256, Beyotime) overnight at 4 °C. The detailed information on the antibodies is included in Appendix A. After three washes in TBST (150 mM NaCl, 20 mM Tris-HCl, pH8.0, 0.1% Tween), the membranes were incubated with secondary antibody for 1 h at RT. Proteins were detected by chemiluminescence using ECL kit (GE2301, GENVIEW).

### 4.3. Fluorescence In Situ Hybridization (FISH)

FISH was conducted pursuant to the following method. Briefly, cells grown on coverslips were treated with 4% paraformaldehyde (PFA) and permeabilization solution (5% Triton-X, 20 mM HEPES pH7.5, 50 mM NaCl, 3 mM MgCl_2_, 300 mM sucrose). After the last wash by PBS, coverslips were dehydrated consecutively in 70%, 95%, and 100% ethanol for 5min each. FITC labeled PNA probe (PANAGENE, sequences in Appendix A) hybridizing solution (70% formamide, 10% blocking reagent (Roche, Basel, Switzerland), 1M Tris-HCl pH7.4, Buffer MgCl_2_ (25 mM MgCl_2_, 9 mM Citric Acid, 82 mM Na_2_HPO_4_)) was added to the coverslips for denaturing at 80 °C for 5 min, and they were hybridized for 2 h at RT in the dark. The coverslips were washed with Wash I solution (70% formamide, 10 mM Tris-HCl, pH7.0–7.5) and Wash II solution (0.15M NaCl, 100 mM Tris-HCl, pH7.0–7.5, 0.08% Tween) for 15 min each, and they were dehydrated consecutively in 70%, 95%, and 100% ethanol for 5 min each. The cells were counterstained with DAPI (H1800, VECTASHIED), and digital images were captured by fluorescence microscope (Leica, DM3000, Wetzlar, Germany).

### 4.4. Hematoxylin and Eosin (H&E) Staining

Testis sections were dewaxed at 65 °C for 30 min, and they were incubated with xylene I, II for 10 min and anhydrous ethanol for 5 min twice. The tissues were dehydrated in 90%, 80%, and 70% ethanol and then washed three times with distilled water and incubated with hematoxylin for 15 min. After washes with the distilled water, the tissues were differentiated 20 s by 1% ethanol hydrochloride and stained with eosin. After two rounds of dehydration with xylene for 4 min, and the sections were sealed with neutral gum.

### 4.5. Immunohistochemistry

Paraffin-embedded testis tissues were cut into 4 μm thick sections, and they were processed via dewaxing, dehydration, antigen retrieval, and 5% BSA blocking. These sections were incubated with primary antibodies (1:50), including anti-ZBTB40, anti-ki67, anti-Vasa, and anti-γH2AX, overnight at 4 °C and incubated with Alexa fluor 488 donkey anti-rabbit lgG or Alexa fluor 594 donkey anti-mouse lgG for 1 h at RT. Cell nuclei were stained with DAPI, and the images were captured with a positive fluorescence microscope (Leika, DM3000, Wetzlar, Germany).

### 4.6. TUNEL Assay

TUNEL apoptosis Detection Kit (Servicebio, Wuhan, China) was used to measure apoptosis. According to the manufacturer’s instructions, testis sections were processed via dewaxing and dehydration, and they were incubated with 20 μg/mL protease K for 30 min at room temperature. TMR-5-dUTP Labeling/TdT enzyme buffer was used to detect the apoptotic cells, and nuclei were stained with DAPI. The images were captured with a fluorescence microscope (Leika, DM3000, Wetzlar, Germany).

### 4.7. Isolation of Mouse Male Germ Cells

Mouse male germ cells were isolated by two-step enzymatic digestion, followed by differential plating. The testicular tissues from *Zbtb40* KO mice and the wild-type (WT) mice were minced and incubated with 10 mL DMEM containing 2 mg/mL type IV collagenase (Gibco) and 10 μg/mL DNase I (Sigma, Saint Louis, MO, USA) for 10 min. Natural settlement for 10 min was performed to obtain the seminiferous tubules that were incubated with 10 mL DMEM, including 2.5 mg/mL type IV collagenase, 2 mg/mL hyaluronidase (Sigma, Saint Louis, MO, USA), 2 mg/mL trypsin (Sigma, Saint Louis, MO, USA), and 10 μg/mL DNase I for 5 min. Cell suspension was collected and filtered through a 40 µm nylon mesh to remove cell aggregates. The cell mixture was seeded in a 10 cm dish pre-coated with gelatin and cultured at 34 °C in a 5% CO_2_ incubator. After culturing for 6 h, the suspension containing male germ cells was collected, and other somatic cells were attached to the cell dish.

### 4.8. STA-PUT Separation of Mouse Spermatocytes

Mouse primary germ cells were isolated by two-step enzymatic digestion and differential adhesion. After culturing for 6 h, cell suspension was collected, and centrifuging was performed at 1500 rpm for 5 min. The cells were resuspended with 0.5% BSA. STA-PUT instrument was added with 300 mL of 4% BSA, 300 mL of 2% BSA, and 50 mL of 0.5% BSA. Twenty-five milliliters of cell suspension were added into the settling tank. BSA at 0.5%, 2%, and 4% concentrations were connected for 3 h to establish a serial of gradient concentrations of BSA to allow male germ cells of different weights and sizes to form cell layers. Afterwards, the cells were collected with 10 mL of cell suspension per tube, and 45 tubes were utilized. Cell morphology in each tube was observed under a microscope.

### 4.9. Spermatocyte Spreading and Immunofluorescence Staining

A mouse testicular sample was incubated in hypotonic extraction buffer (30 mM tris (pH 8.2), 50 mM sucrose, 17 mM citric acid, 5 mM EDTA, 2.5 mM dithiothreitol, and 1 mM phenylmethylsulfonyl fluoride) for 30 min at RT. Subsequently, cells were released by squeezing with toothless tweezers in 100 mM sucrose. Cell suspension (10 μL per slide) was spread on slides with 1% PFA (pH 9.2) containing 0.15% Triton X-100, and cell slides were allowed to dry overnight in the humid chamber at RT. Then, slides were washed with 0.04% Photo-Flo (KODAK Cat.NO.1464510) for 4 min and air-dried at RT for 10 min. The primary antibodies included SYCP3 (1:3000, abcam, Ab15093) and γ-H2AX (1:3000, millipore, 05-636) and were incubated overnight at 37 °C. Slides were washed 3 times with ADB (1% normal donkey serum (Jackson ImmunoResearch, 017-000-121), 2.5% BSA (Roche, Basel, Switzerland, Cat.NO.100350), and 0.005% triton-X-100), and they were incubated with ADB overnight at 4 °C. The second antibodies were donkey anti-mouse 488 and donkey anti-rabbit 555, and they were incubated with cell slides for 90 min at 37 °C. Finally, images were captured by a fluorescent microscope (Leica, Wetzlar, Germany, DM3000).

### 4.10. Sperm Counting

Mouse epididymis was isolated and cut with ophthalmological scissors, and 1ml of G-IVF^TM^ plus (Vitrolife, Gothenburg, Sweden, LOT509784) was added to it and incubated at 37 °C for 30 min. Suspension was collected and centrifuged at 300× *g* for 5 min, while 2 mL G-IVF^TM^ plus were used to resuspend sperm. In total, 10 μL sperm suspension was utilized for counting. The sperm suspension was employed for Pap staining and immunofluorescence or fixed with glutaraldehyde for transmission electron microscopy (TEM).

### 4.11. Transmission Electron Microscope (TEM)

Sperm was prefixed with a 2.5% glutaraldehyde, postfixed in 1% osmium tetroxide, dehydrated in series acetone, infiltrated in Epox 812, and embedded. The semithin sections were stained with methylene blue, and ultrathin sections were cut with a diamond knife and stained with uranyl acetate and lead citrate. Sections were examined under a JEM-1400-FLASH TEM.

### 4.12. Pap Staining

Sperm smear was fixed with an equal mixture of 95% ethanol and ether for 15 min, and it was dehydrated in 80%, 70%, and 50% ethanol for 2 min, respectively. The slides were washed twice with distilled water and stained with hematoxylin for 3 min, and Scott’s solution was used for 4 min. The sperm smear was dehydrated again with the graded ethanol, and it was processed with orange-yellow G6 dye solution for 2 min, 95% ethanol twice for 2 min, EA50 staining solution for 5 min, 95% ethanol three times for 1 min each, absolute ethanol for 2 min, and xylene transparent three times for 1 min.

### 4.13. Statistical Analysis

The *t*-test or single factor ANOVA analysis was performed using the SPSS statistical software and GraphPad Prism. Three independent replicates were performed for each experiment, and the data were expressed as mean ± SD. The Student’s *t*-test was employed for comparison between the two groups. The *p* < 0.05 values indicated that the difference was statistically significant.

## Figures and Tables

**Figure 1 cells-12-01264-f001:**
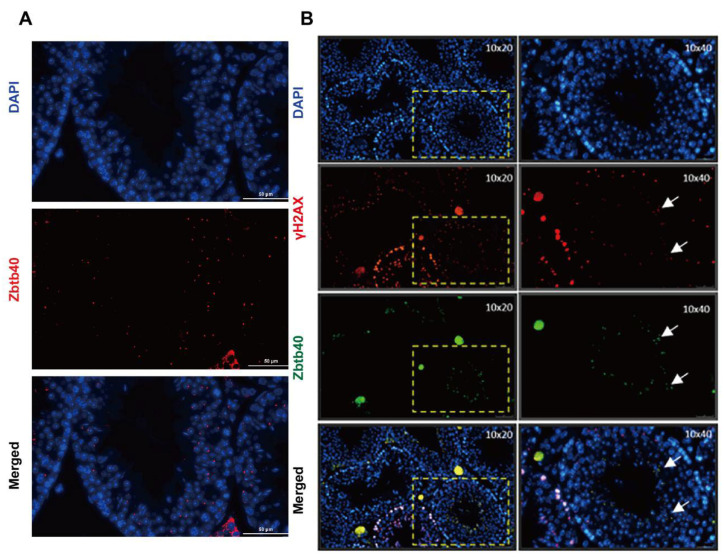
The expression of ZBTB40 in mouse testes. (**A**) Immunohistochemistry showed the expression of Zbtb40 in mouse spermatocytes. (**B**) Double immunofluorescence revealed the co-localization of Zbtb40 with γH2AX in mouse spermatocytes. Right panel is the enlargement of areas in left panel. White arrows indicated the spermatocytes that were positive for Zbtb40 and γH2AX.

**Figure 2 cells-12-01264-f002:**
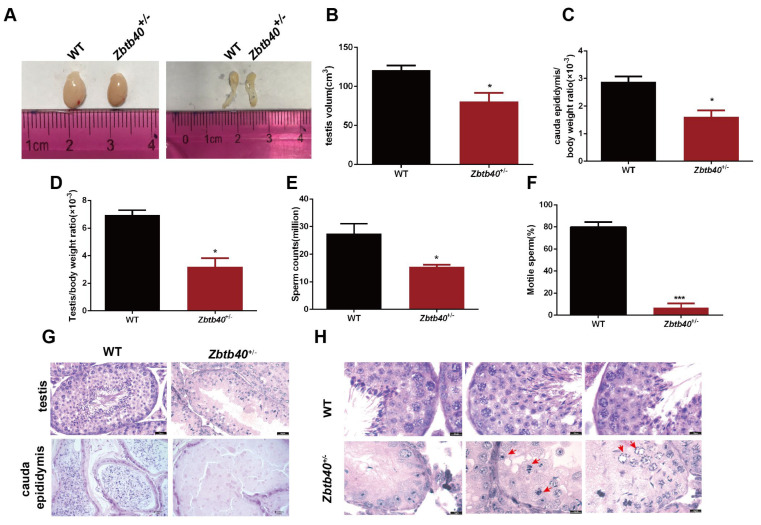
The phenotypical characteristics of *Zbtb40*^+/−^ male mice. (**A**) Testes (left panel) and cauda epididymis (right panel) of the *Zbtb40*^+/−^#4 mice and the WT mice. (**B**–**D**) Comparison of testicular volume (**B**), cauda epididymis and body weight ratio (**C**), and testis/body weight ratio (**D**) of the 5-month-old *Zbtb40^+/^*^−^ male mice and the WT mice. (**E**) Sperm counts were reduced in the *Zbtb40*^+/−^ mice compared to the WT mice. (**F**) Motile sperm was decreased in the *Zbtb40*^+/−^ mice compared to the WT mice. For *p* values in (**B**–**F**) * indicated *p* < 0.05, *** implicated *p* < 0.001, *n* = 3. (**G**) H&E staining demonstrated abnormal morphology of testes and cauda epididymis of the *Zbtb40*^+/−^ mice. (**H**) Representative pictures of testes in the *Zbtb40*^+/−^ mice and WT mice. Red arrows indicated pyknotic spermatocytes.

**Figure 3 cells-12-01264-f003:**
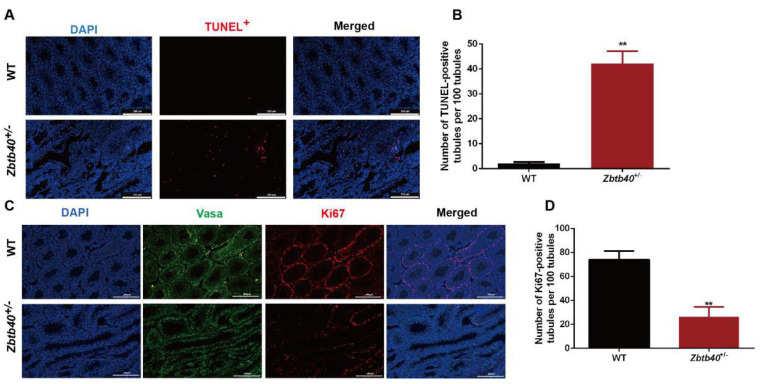
The influence of *Zbtb40*^+/−^ on apoptosis and the proliferation of mouse male germ cells. (**A**,**B**) TUNEL measured percentages of apoptosis between *Zbtb40*^+/−^ mice and the WT mice. (**C**,**D**) Immunohistochemistry showed Ki67-positive germ cells between *Zbtb40*^+/−^ mice and the WT mice. For *p* values of B and D: ** denoted *p* < 0.01, *n* = 3.

**Figure 4 cells-12-01264-f004:**
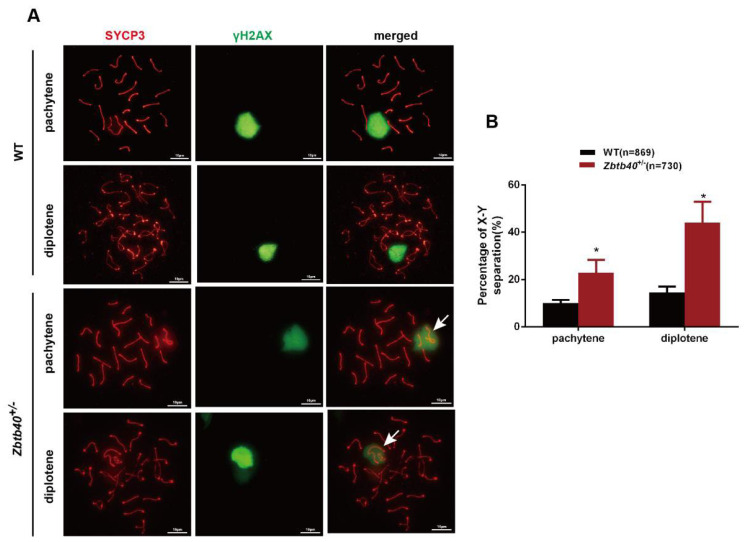
The defective phenotypes of spermatocytes of *Zbtb40*^+/−^ mice. (**A**) Representative spermatocyte spreading revealed the abnormal XY dissociation in *Zbtb40*^+/−^(−400bp) mice and the WT mice. The white arrows denote sex chromosome. (**B**) The bar chart presents the rates of XY separation in samples of *Zbtb40*^+/−^ mice and the WT mice. For *p* values in B * indicated *p* < 0.05.

**Figure 5 cells-12-01264-f005:**
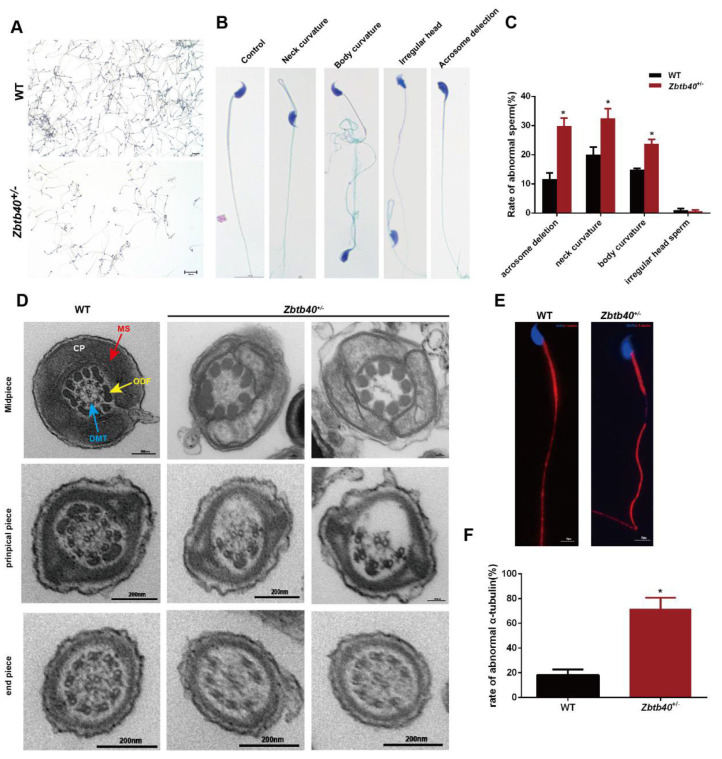
The sperm of *Zbtb40*^+/−^ male mice show flagellum deformities. (**A**) Pap stain was used to observe sperm morphology of the *Zbtb40*^+/−^ mice and the WT mice. (**B**) Representative photos of abnormal sperm of the *Zbtb40*^+/−^ mice. (**C**) The bar chart indicates the rates of abnormal sperm in samples of the *Zbtb40*^+/−^ mice. (**D**) TEM of the cross-sections of sperm flagella displayed abnormal axoneme and peri-axoneme structures of the *Zbtb40^+/^*^−^ mice. ODFs: outer dense fibers; DMT: outer doublet microtubules; MS: mitochondrial sheath; CP: central pair complex. (**E**) Immunofluorescence was used to observe expression of α-tubulin in sperm tails of the Zbtb40^+/−^ mice. (**F**) The bar chart shows the rates of abnormal α-tubulin in the *Zbtb40*^+/−^ mice. For *p* values of (**C**,**F**): * indicated *p* < 0.05, *n* = 3.

**Figure 6 cells-12-01264-f006:**
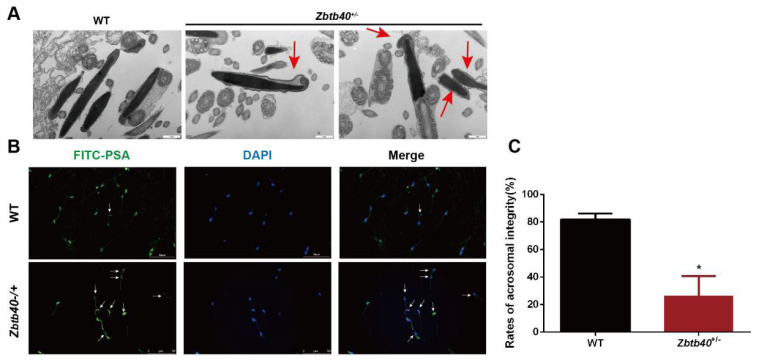
The sperm of *Zbtb40*^+/−^ male mice assume abnormal acrosome biogenesis. (**A**) TEM was used to detect the acrosome biogenesis in the *Zbtb40*^+/−^ mouse. Red arrows indicate abnormal acrosome. (**B**) Immunofluorescence was used to detect the expression of FITC-PSA in sperm head of the *Zbtb40*^+/−^ mice. The white arrows denote flawed acrosome. (**C**) The bar chart displays the rates of acrosomal integrity in the *Zbtb40*^+/*−*^ mice. For *p* values of *t*-tests in (**C**): * indicated *p* < 0.05, *n* = 3.

**Figure 7 cells-12-01264-f007:**
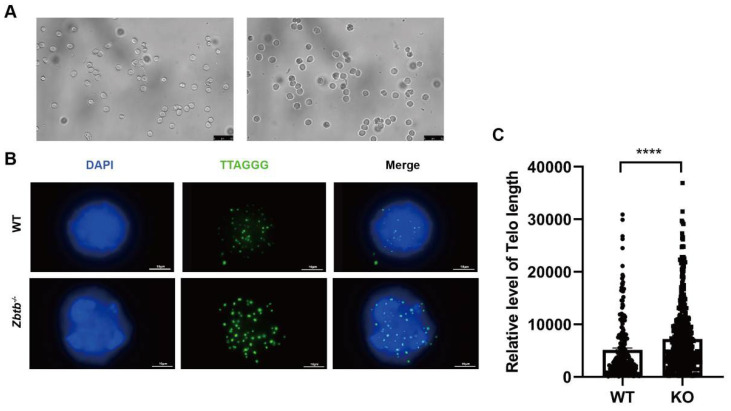
*Zbtb40*^+/−^ mice have longer telomere length. (**A**) Representative spermatocytes were separated by STA-PUT from the *Zbtb40*^+/−^ mice. (**B**) Q-FISH showed telomere foci of spermatocytes in the *Zbtb40*^+/−^ mice and the WT mice. (**C**) The distribution of relative telomere length in the *Zbtb40*^+/−^ mice and the WT mice. For *p* values of *t*-tests in (**C**): **** denoted *p* < 0.0001.

## Data Availability

The data could be available with the consent of the corresponding author.

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
