# Peer review of "Zbtb40 Deficiency Leads to Morphological and Phenotypic Abnormalities of Spermatocytes and Spermatozoa and Causes Male Infertility"

_cells, 2023, doi:10.3390/cells12091264_

Round 1

Reviewer 1 Report

1. In 2.1, colocalization in spermatocytes was not so obvious. The cell on the lefthand side had a stronger red but not very much green, from what I could see from the photo. It would be better if the authors provide more convincing evidence.

2. In 2.2, the article stated that KO and WT mice had similar outlooks, but with different fertility. It would be more convincing to enclose photos of the mice, and offspring numbers of different matings. Also, there was no Fig2I. Please check your writings.

3. In 2.6, the article stated that KO mice had longer telomeres. However, longer telomeres are generally concerned as a sign of healthier and younger cells, and there has been research suggesting shorter telomeres in white blood cells in NOA patients and associated with low spermatogenesis. Could the authors give a more detailed explanation how longer telomeres in mutant spermatocytes might be related to abnormal spermatogenesis?

4. English grammar and accurate wording should be checked thoroughly. For example, in 2.4,  Unlike autosomes, there was a very 134 short but highly homologous DNA sequences..., the word sequenceshould be in singular form, was should be is as this sentence is stating a clear fact, and 134 just should not be where it is. 

Author Response

  1. In 2.1, colocalization in spermatocytes was not so obvious. The cell on the lefthand side had a stronger red but not very much green, from what I could see from the photo. It would be better if the authors provide more convincing evidence.

Response:Thank you for your suggestion. We have now provided new data showing the colocation of ZBTB40 and rH2AX in new Figure 1.

  1. In 2.2, the article stated that KO and WT mice had similar outlooks, but with different fertility. It would be more convincing to enclose photos of the mice, and offspring numbers of different matings. Also, there was no Fig2I. Please check your writings.

Response:Thanks for your comments. We have enclosed the photos of the WT mice and ZBTB40-/- mice in new Figure S1B, and we have included the number of offspring in new Figure S1C. We have removed Fig2H.

  1. In 2.6, the article stated that KO mice had longer telomeres. However, longer telomeres are generally concerned as a sign of healthier and younger cells, and there has been research suggesting shorter telomeres in white blood cells in NOA patients and associated with low spermatogenesis. Could the authors give a more detailed explanation how longer telomeres in mutant spermatocytes might be related to abnormal spermatogenesis?

Response:In this study, we found that ZBTB40-KO mice have longer telomere in spermatocytes. This might be due to the highly heterogeneous male germline telomere length (Journal of Assisted Reproduction and Genetics, 2015, 32: 1685-1690). It has been reported that the on-average longer telomeres in older men are due to subpopulations of sperm with longer telomeres , further highlighting the heterogeneity in sperm (PLoS Genetics, 2008, 4: e37). Disruption of telomerase activity or telomere length timing may also cause male infertility (Cell Reports, 2021, 37(11):110110). 

  1. English grammar and accurate wording should be checked thoroughly. For example, in 2.4,  “Unlike autosomes, there was a very 134 short but highly homologous DNA sequences...”, the word “sequence” should be in singular form, “was” should be “is” as this sentence is stating a clear fact, and “134” just should not be where it is. 

Response:Thanks for your suggestion. we have made correction on ‘is’ and checked the manuscript thoroughly to ensure no grammar or spelling error.

Reviewer 2 Report

 Dear Editor and authors,

Thank you for the opportunity to review the manuscript titled " Zbtb40 deficiency leads to morphological and phenotypic abnormalities of spermatocytes and spermazoa and causes male infertility" After careful consideration of the manuscript, I have found the study to be compelling and significant.

The authors have employed a logical workflow and utilized various techniques to validate their findings, which is commendable. The results of their study provide novel insights into the molecular mechanisms underlying male infertility, and the involvement of Zbtb40 in this process. The observations of morphological and phenotypic abnormalities of spermatocytes and spermatids, as well as telomere dysfunction in Zbtb40-deficient mice, are noteworthy and contribute to the current understanding of male infertility.

Furthermore, the manuscript is well-written and organized, making it easy for readers to follow and understand the study's results and conclusions. The authors have also provided ample details on the methodology and statistical analyses used, which adds to the manuscript's credibility.

However, there is a few important caveats is that since this study involves some statistical analyses, the data should be available in a data depository centres, or alternatively all test descriptive including test results, mean, sd, df, sample size etc should be added to supplementary tables. In addition, the reference can be improved. There is an in-detail discussion but relatively a few studies were mentioned.

In summary, I recommend this manuscript for publication in your journal when data is provided, given its novel findings, logical workflow, and well-written presentation.

Below I listed a minor mistake, and suggested some improvements.

Thank you for considering my feedback.

Sincerely,

 BP.

Title (also in Supp data) – “spermatozoa” instead of “spermazoa”

Line 17 and 62: Italic Zbtb40

Line 34: mammals

Line: 37-38: suggested edition: “Abnormal spermatogenesis can lead to male infertility, affecting about 15% of couples worldwide.”

Be sure the nomenclator of gene and mutation names are followed (MGI-Guidelines for Nomenclature of Genes, Genetic Markers, Alleles, & Mutations in Mouse & Rat (https://www.informatics.jax.org/mgihome/nomen/gene.shtml))

Line 73-74: It is not clear the aim here. You looked at the gene ortholog and found in the mouse after E. coioides study. That’s the first thing came to my mind when I read nut indeed this paragraph is a immunostaining study, which is nicely presented in the figure but this paragraph can be improved by adding the aim and methods with a few words.

Line 86: Says Zbtb40 KO mice 85 (-400bp or -391bp, Fig. S1) that refers Zbtb40+/- #4 and Zbtb40+/- #98 respectively. Figure S2-C shows Zbtb40-/- and Zbtb40+/- (-300 bp and -400 bp). I could have missed some details, otherwise please check is it the same samples were mentioned here.

Line 93: males assumed? You meant showed/were?

Figure 2 and others: No sample size were mentioned in the results and material methods,  please add this information. What are the error bars represented for? Sd, se or ci?

Line 220: “the” longer telomeres (no article needed)

Line 221: patients

Line 223: disease-causing

Author Response

Dear Editor and authors,

Thank you for the opportunity to review the manuscript titled " Zbtb40 deficiency leads to morphological and phenotypic abnormalities of spermatocytes and spermazoa and causes male infertility" After careful consideration of the manuscript, I have found the study to be compelling and significant.

The authors have employed a logical workflow and utilized various techniques to validate their findings, which is commendable. The results of their study provide novel insights into the molecular mechanisms underlying male infertility, and the involvement of Zbtb40 in this process. The observations of morphological and phenotypic abnormalities of spermatocytes and spermatids, as well as telomere dysfunction in Zbtb40-deficient mice, are noteworthy and contribute to the current understanding of male infertility.

Furthermore, the manuscript is well-written and organized, making it easy for readers to follow and understand the study's results and conclusions. The authors have also provided ample details on the methodology and statistical analyses used, which adds to the manuscript's credibility.

However, there is a few important caveats is that since this study involves some statistical analyses, the data should be available in a data depository centres, or alternatively all test descriptive including test results, mean, sd, df, sample size etc should be added to supplementary tables. In addition, the reference can be improved. There is an in-detail discussion but relatively a few studies were mentioned.

In summary, I recommend this manuscript for publication in your journal when data is provided, given its novel findings, logical workflow, and well-written presentation.

Below I listed a minor mistake, and suggested some improvements.

Thank you for considering my feedback.

Sincerely,

 BP.

Title (also in Supp data) – “spermatozoa” instead of “spermazoa”

Line 17 and 62: Italic Zbtb40

Line 34: mammals

Line: 37-38: suggested edition: “Abnormal spermatogenesis can lead to male infertility, affecting about 15% of couples worldwide.”

Be sure the nomenclator of gene and mutation names are followed (MGI-Guidelines for Nomenclature of Genes, Genetic Markers, Alleles, & Mutations in Mouse & Rat (https://www.informatics.jax.org/mgihome/nomen/gene.shtml))

Line 73-74: It is not clear the aim here. You looked at the gene ortholog and found in the mouse after E. coioides study. That’s the first thing came to my mind when I read nut indeed this paragraph is a immunostaining study, which is nicely presented in the figure but this paragraph can be improved by adding the aim and methods with a few words.

Line 86: Says Zbtb40 KO mice 85 (-400bp or -391bp, Fig. S1) that refers Zbtb40+/- #4 and Zbtb40+/- #98 respectively. Figure S2-C shows Zbtb40-/- and Zbtb40+/- (-300 bp and -400 bp). I could have missed some details, otherwise please check is it the same samples were mentioned here.

Line 93: males assumed? You meant showed/were?

Figure 2 and others: No sample size were mentioned in the results and material methods,  please add this information. What are the error bars represented for? Sd, se or ci?

Line 220: “the” longer telomeres (no article needed)

Line 221: patients

Line 223: disease-causing

Response:We thank the reviewers for the positive comments that our study provides novel insights into the molecular mechanisms underlying male infertility and our study is noteworthy and contributes to the current understanding of male infertility.

We have described the statistics and provided the values along with p values for the parameters measured in our revised manuscript. We have checked our manuscript thoroughly, and we have made all corrections on the above mistaken in the revised manuscript.

Reviewer 3 Report

The manuscript is in general well-written and interesting. Please consider the following suggestions to improve the paper:There are small grammatical errors in several areas of the manuscript. Consider having the manuscript edited for English.

Introduction

Line 38: Consider describing non-obstructive azoospermia a bit more in the introduction.

Lines 40-41: Consider adding a sentence connecting asthenoteratospermis to NOA.

Results

Lines 92-93, 99, and 102, and elsewhere in the manuscript: please provide values along with p values for the parameters measured when describing a variable as being “remarkably reduced”, “lower” or “difference”

MethodsPlease provide the number of mice used for this experiment in the methods and in the figure descriptions.

Be sure to include the “n” in the figure descriptions.

In general, the statistics need to be better described. Consider stating which comparisons were made within each section of the methods.

Author Response

The manuscript is in general well-written and interesting. Please consider the following suggestions to improve the paper:

There are small grammatical errors in several areas of the manuscript. Consider having the manuscript edited for English.

Introduction

Line 38: Consider describing non-obstructive azoospermia a bit more in the introduction.

Lines 40-41: Consider adding a sentence connecting asthenoteratospermis to NOA.

Response:Thanks for your suggestion. We have provided more detained information on non-obstructive azoospermia and asthenoteratospermis.

Results

Lines 92-93, 99, and 102, and elsewhere in the manuscript: please provide values along with p values for the parameters measured when describing a variable as being “remarkably reduced”, “lower” or “difference”

Response:We have provided values along with p values in our revised manuscript.

Methods
Please provide the number of mice used for this experiment in the methods and in the figure descriptions.

Be sure to include the “n” in the figure descriptions.

In general, the statistics need to be better described. Consider stating which comparisons were made within each section of the methods.

Response:We have indicated in the statistical analysis. And the “n” has been included in the results or Figure legends.

Round 2

Reviewer 2 Report

I am writing to express my gratitude for taking the time to consider my suggestions for your manuscript. I was delighted to see that my feedback was taken into account, and that the revised version is significantly improved.

Author Response

We appreciate that this reviewer was satisfied with our revised manuscript and that our revised version has been significantly improved.

Reviewer 3 Report

The manuscript is much improved. Prior to publishing, please consider the following:

Introduction

Please include objectives and specific aims of the study, along with a hypothesis.

Line 68: remove the word “specially”, and instead use “specifically”

Lines 68-76: This paragraph is better suited for the discussion section.

Author Response

The manuscript is much improved. Prior to publishing, please consider the following:

We are grateful to this reviewer for the nice comment that our revised manuscript has beent much improved. We have further modified our manuscript in terms of the helpfully suggestion as follows:

 Introduction

Please include objectives and specific aims of the study, along with a hypothesis.

Lines 69-92, we have now included the aim and a hypothesis of this study: The aim of this study was designed to explore the expression and role of Zbtb40 in mediating male reproduction and male fertility. To this end, we hypothesized that the loss of ZBTB40 function may result in the abnormal spermatogenesis and male infertility.

Line 68: remove the word “specially”, and instead use “specifically”

We have changed word “specially” to “specifically”, line 291.

Lines 68-76: This paragraph is better suited for the discussion section.

Lines 291-295, We have now moved this paragraph to the section of Discussion: we have found that Zbtb40 is expressed specifically in mouse spermatocytes and that it is co-localized with γH2AX in mouse secondary spermatocytes. Significantly, our loss of function study implicates that the Zbtb40 knockout mice show male infertility and they assume morphologically defective sperm heads, flagella, and acrosome biogenesis.